# HIP1R Expression and Its Association with PD-1 Pathway Blockade Response in Refractory Advanced NonSmall Cell Lung Cancer: A Gene Set Enrichment Analysis

**DOI:** 10.3390/jcm9051425

**Published:** 2020-05-11

**Authors:** Young Wha Koh, Jae-Ho Han, Seokjin Haam, Hyun Woo Lee

**Affiliations:** 1Department of Pathology, Ajou University School of Medicine, Suwon 16499, Korea; hanpathol@naver.com; 2Department of Thoracic and Cardiovascular Surgery, Ajou University School of Medicine, Suwon 16499, Korea; haamsj@aumc.ac.kr; 3Department of Hematology-Oncology, Ajou University School of Medicine, Suwon 16499, Korea; leehw@ajou.ac.kr

**Keywords:** nonsmall cell lung cancer, HIP1R, PD-L1, biomarker

## Abstract

Huntingtin-interacting protein 1-related protein (HIP1R) plays an important role in the regulation of programmed death-ligand 1 (PD-L1). The aim of this study was to investigate the expression of HIP1R and confirm its predictive or prognostic roles in anti-PD-1 therapy in nonsmall cell lung cancer (NSCLC) patients. HIP1R and PD-L1 immunohistochemical expression was examined in 52 refractory advanced NSCLC patients treated with anti-PD-1 inhibitors. We performed gene set enrichment analysis (GSEA) to detect HIP1R-specific gene sets. Patients in the PD-1 inhibitor responder group had lower HIP1R expression by univariate logistic regression analysis (odds ratio (OR) = 0.235, *p* = 0.015) and multivariate logistic regression analysis (OR = 0.209, *p* = 0.014). Patients with high HIP1R expression had poorer progression-free survival (PFS) than patients with low HIP1R expression in univariate analysis (*p* = 0.037) and multivariate Cox analysis (hazard ratio = 2.098, *p* = 0.019). The web-based mRNA dataset also showed that high HIP1R expression correlated with inferior overall survival in lung adenocarcinoma (*p* = 0.026). GSEA revealed that HIP1R levels correlate with a set of genes that reflect PD-L1-related immune pathways. HIP1R expression may be a promising predictor for determination of patient responses to anti-PD-1 treatment.

## 1. Introduction

Emergence of immune checkpoint inhibitors was a turning point in the treatment of advanced nonsmall cell lung cancer (NSCLC). Therapies targeting the programmed cell death protein 1 (PD-1) checkpoint, such as nivolumab and pembrolizumab, have yielded impressive responsive rates in advanced NSCLC patients otherwise refractory to multiples lines of therapy [1,2,3]. However, the overall response rate for PD-1 inhibitor therapy is approximately 15–20% in unselected patients with NSCLC, and between 15% and 45% in patients with PD-L1-expressing NSCLC [4]. We need a biomarker that can more accurately predict the response to PD-1 inhibitors.

Expression of programmed death ligand-1 (PD-L1), the PD-1 ligand, is currently the most widely used biomarker for PD-1 inhibition. To identify patients who preferentially respond to PD-1 blockade, we need to better understand how the PD-1 pathway is regulated. Recently, several mechanisms have been reported to underlie PD-1 pathway regulation. CKLF-like MARVEL transmembrane-domain-containing 6 (CMTM6) regulates the PD-1 pathway by maintaining the expression of PD-L1, and CMTM6 is a predictor of the response to PD-1 inhibitors [5,6]. F-box only protein 38 (FBXO38) mediates PD-1 ubiquitination of T cells, and knockout of FBXO38 in such cells induces tumor progression in a mouse model due to increased PD-1 expression by tumor-infiltrating T cells [7]. AXL expression displays a positive correlation with PD-L1 expression in lung adenocarcinoma with epidermal growth factor receptor (EGFR) mutation, and abolition of AXL kinase activity inhibits *PD-L1* mRNA expression in a lung adenocarcinoma cell line with EGFR mutation [8].

Recent research has uncovered new strategies to remove specific unwanted proteins by using cellular protein degradation mechanisms, including lysosome-targeting molecules [9], proteolysis-targeting chimeras (PROTACs) [10], and tag-based degradation systems (dTAG) [11]. Wang et al. reported that Huntingtin-interacting protein 1-related protein (HIP1R) promotes lysosomal degradation of PD-L1, inhibits HIP1R-induced PD-L1 accumulation, and alters T cell–mediated cytotoxicity in a human colorectal cancer cell line [12]. A chimeric peptide including a lysosomal sorting signal and the HIP1R PD-L1-binding sequence significantly inhibits PD-L1 protein expression [12]. Although immune checkpoint inhibition is the most popular treatment for lung cancer, relationships involving HIP1R and immune checkpoint inhibitors in lung cancer have not been studied.

The present study was conducted to determine whether HIP1R protein expression affects the response of NSCLC patients to anti-PD-1 inhibitors and their prognosis. The relationship between HIP1R and PD-L1 was also evaluated, employing immunohistochemical and web-based mRNA expression data. In addition, we performed gene set enrichment analysis (GSEA) on RNA-sequencing data from The Cancer Genome Atlas (TCGA) to confirm the molecular pathways associated with HIP1R expression.

## 2. Materials and Methods

### 2.1. Patients

We retrospectively selected 52 advanced NSCLC patients who were administered PD-1 inhibitor from 2016 to 2019, and they previously received one or two lines of chemotherapy. This study was approved by the Institutional Review Board of Ajou University School of Medicine. Informed consent was waived due to the retrospective nature of the study (AJIRB-BMR-KSP-19-050 and 2019-03-26).

Patients treated with PD-1 inhibitor were assigned to either a responder group (complete response, partial response, or stable disease) or a nonresponder group (disease progression), according to the response evaluation criteria for solid tumors (RECIST) version 1.1 [13].

### 2.2. Immunohistochemical Staining and HIP1R Expression Scoring

One board-certified pathologist (YWK) reviewed hematoxylin and eosin (H&E)-stained tissue samples to determine a definitive pathologic diagnosis according to the 2015 World Health Organization Classification of Lung Tumors [14]. All patients were pathologically staged according to the eighth edition of the TNM classification.

HIP1R immunohistochemical (IHC) staining was performed with a Benchmark XT automatic IHC staining device (Ventana Medical Systems, Tucson, AZ, USA). The samples were incubated with an anti-HIP1R antibody (dilution 1:1000, 16814-1-AP, polyclonal, Proteintech, Rosemont, IL, USA). We used a human placenta tissue as positive control according to the manufacturer’s recommendations (Appendix A). We also evaluated the intensity of HIP1R staining on a four-point intensity scale: 0 (no staining), 1 (light yellow = faint staining), 2 (yellow-brown = moderate staining), and 3 (brown = strong staining) (Figure 1). We also evaluated the percentages (0–100%) of cytoplasmic versus membranous localization of HIP1R. We used H-scores to interpret HIP1R staining [15], where H-score = [1 × (% cells 1+) + 2 × (% cells 2+) + 3 × (% cells 3+)]. H-scores (0–300) were obtained by multiplying the percentage of cells by the intensity score.

### 2.3. Immunohistochemical Staining and PD-L1 Expression Scoring

Two PD-L1 antibodies (clone name SP263 or 22C3) were used to detect PD-L1 expression. Sp263 was a companion diagnostic assay for OPDIVO^®^ (nivolumab), and 22c3 was a companion diagnostic assay for KEYTRUDA^®^ (pembrolizumab). We performed SP263 and/or 22C3 assays prior to PD-1 inhibitor treatment for all NSCLC patients. Thirteen (25%) of the 52 specimens were tested for both SP263 and 22C3, 27 (51.9%) for only SP263, and 12 (23.1%) for only 22C3. Two PD-L1 tests used prediluted antibody (ready to use) according to the protocol. The SP263 assay was performed using a VENTANA BenchMark ULTRA instrument (Roche, Basel, Switzerland), and the 22C3 assay was conducted using the Dako Link-48 platform (Dako, Carpinteria, California, US), as recommended by the manufacturers [16]. PD-L1 intensity was also evaluated on a four-point intensity scale (0, none; 1, faint; 2, moderate; and 3, strong), and the percentage of membranous expression of PD-L1 was determined (Appendix A). When both the 22C3 and SP263 tests were conducted, mean values were used. High PD-L1 expression was defined as ≥ 50% of definitive tumor cells exhibiting PD-L1 staining, because 50% was the cut-off used for NSCLC [17].

### 2.4. Web-Based mRNA Profiling, GSEA, and Kaplan Meier Analysis

The mRNA sequencing data of 517 lung adenocarcinoma patients and 501 lung squamous cell carcinoma patients were downloaded from The Cancer Genome Atlas (TCGA) cBioportal (http://cbioportal.org) [18]. We conducted correlation analysis involving PD-L1 and HIP1R mRNA sequencing data. 

GSEA is a method of analyzing associations between gene expression and biological information. We conducted GSEA using GSEA version 4.0.3 from the Broad Institute at MIT and Harvard (http://www.broadinstitute.org/gsea/index.jsp) [19]. TCGA mRNA sequencing data derived from lung adenocarcinoma and lung squamous cell carcinoma patients was used. Depending on the median value, it is divided into low and high HIP1R. Hallmark gene sets representing well-defined biological states or processes were used for GSEA. 1000 permutations were used for estimating nominal *p* values. If the *p* value was less than 0.05 and the False Discovery Rate (FDR) was less than 0.25, the findings were considered statistically significant.

We conducted survival analyses using an online Kaplan Meier plotter tool [20]. The online Kaplan Meier plotter tool provides mRNA expression data of cancer patients and allows for survival analysis. Survival analyses were performed in 719 patients with lung adenocarcinoma and 524 lung squamous cell carcinoma cases according to their HIP1R mRNA expression.

### 2.5. Statistical Analyses

Spearman’s rank order correlation coefficient analysis was used to measure monotonic relationships between continuous variables. Mann Whitney U-tests were used to compare differences between two independent groups. Univariate and multivariate logistic regression analyses were performed to determine factors that predicted a response to PD-1 inhibitors. Receiver operating curve (ROC) analysis was used to determine the cut-off values for HIP1R expression. The progression-free survival (PFS) or overall survival (OS) difference between the cohorts was determined using the log-rank test. Univariate and multivariate prognostic analyses were performed for PFS using a Cox proportional hazards regression model. SPSS Statistics for Windows (Version 25.0, IBM, Armonk, NY, USA) was used for all analyses, and *p* values <0.05 were considered to be statistically significant.

## 3. Results

### 3.1. Patient Demographics

Detailed patient and tumor characteristics are summarized in Table 1. Forty-six tissues were collected from lung lesions, and six were obtained from metastatic sites. Twenty-seven (51.9%) patients had been treated with nivolumab, and 25 (48.1%) patients received pembrolizumab. All patients were refractory to conventional treatments such as chemotherapy, radiotherapy, or target therapy. Therefore, they received PD-1 inhibitor as a second line or later setting. Twenty-seven (51.9%) patients were classified as responders, and 25 (48.1%) were classified as nonresponders. Four patients were treated with epidermal growth factor receptor (EGFR) inhibitors before PD-1 inhibitor administration. Patients with anaplastic lymphoma kinase (ALK) fusion were not identified in the present study.

### 3.2. Relationships Between HIP1R and PD-L1 Analyzed by IHC and mRNA Expression

We performed correlation analysis of HIP1R and PD-L1 expression using IHC techniques. There was no statistically significant correlation between HIP1R and PD-L1 expression (*p* = 0.905, Figure 2A).

Correlation analyses of HIP1R and PD-L1 expression were performed using mRNA data. From the TCGA dataset, HIP1R mRNA expression levels were negatively correlated with PD-L1 mRNA levels in adenocarcinoma and squamous cell carcinoma. (Spearman’s rho = −0.233, *p* < 0.001, Figure 2B; Spearman’s rho = −0.224, *p* < 0.001, Figure 2C).

### 3.3. Associations Involving HIP1R, PD-L1, Clinicopathologic Parameters, and Response to PD-1 Inhibitors

ROC analysis was performed to determine the cut-off value for HIP1R expression. Cut-off was determined as the value corresponding to the maximum joint sensitivity and specificity of the ROC curve. The area under the curve (AUC) was 0.659 for the expression of HIP1R, and the cut-off value was 180 (66% sensitivity and 68% specificity, Figure 3).

We explored the predictive capacity of HIP1R, PD-L1, and clinicopathologic factors in terms of responses to PD-1 inhibition. By univariate analysis, the expression of HIP1R was found to be a predictor of the response to anti-PD-1 therapy (OR) = 0.235, *p* = 0.015; Table 2). PD-L1 expression was also found to be a predictor of the response to anti-PD-1 therapy (OR = 4.062, *p* = 0.028; Table 2). By multivariate analysis, the expression of HIP1R was an independent predictor of anti-PD-1 therapy response (OR = 0.209, *p* = 0.014, Table 2).

### 3.4. GSEA According to HIP1R mRNA Expression

We performed GSEA to identify gene sets associated with HIP1R mRNA expression in the TCGA mRNA data of lung adenocarcinoma and lung squamous cell carcinoma cases. In lung adenocarcinoma, we identified the top 20 most prominent pathways that were upregulated in the low HIP1R mRNA expression group (Appendix A). Four of the 20 were immune-related gene sets and were statistically significant (HALLMARK_ALLOGRAFT_REJECTION, HALLMARK_INFLAMMATORY_RESPONSE, HALLMARK_IL6_JAK_STAT3_SIGNALING, and HALLMARK_IL2_STAT5_SIGNALING) (Figure 4). HALLMARK_INTERFERON_GAMMA_RESPONSE is also upregulated in the low HIP1R mRNA expression group, although the statistical significance was marginal (*p* = 0.063). Core enrichment gene lists for HALLMARK_ALLOGRAFT_REJECTION, HALLMARK_INFLAMMATORY_RESPONSE, HALLMARK_IL6_JAK_STAT3_SIGNALING and HALLMARK_IL2_STAT5_SIGNALING are summarized in Appendix A. In lung squamous cell carcinoma, there were no statistically significant immune-related gene sets associated with HIP1R mRNA expression (Appendix A).

### 3.5. Prognostic Significance of HIP1R and PD-L1

Patients with high HIP1R expression had inferior PFS to patients with low HIP1R expression (*p* = 0.037, Figure 5A). Patients with high HIP1R expression also showed an inferior OS than patients with low HIP1R expression, however the statistical significance was not reached (*p* = 0.11, Figure 5B). Patients with high PD-L1 expression had superior PFS or OS than patients with low PD-L1 expression (*p* = 0.028, Figure 5C and *p* = 0.031, Figure 5D, respectively). Furthermore, patients with high HIP1R expression and low PD-L1 expression had lower PFS or OS than patients with other expression patterns (*p* < 0.001, Figure 5E and *p* = 0.001, Figure 5F, respectively). In multivariate analysis, high HIP1R expression was an independent prognostic factor for PFS (HR = 2.098, *p* = 0.019, Table 3).

We used a Kaplan Meier plotter tool and performed survival analysis according to HIP1R mRNA expression in lung adenocarcinoma and lung squamous cell carcinoma patients. The group with high HIP1R mRNA expression exhibited poorer OS in patients with adenocarcinoma (*p* = 0.026, Appendix A). However, in lung squamous cell carcinoma, HIP1R mRNA expression was not correlated with OS (*p* = 0.63, Appendix A)

## 4. Discussion

This study had several novel discoveries. First, we found that the expression of HIP1R was an independent predictive factor for anti-PD-1 treatment response by NSCLC patients. Second, the expression of HIP1R was an independent prognostic factor of PFS in patients treated with anti-PD-1 inhibitors. Third, GSEA revealed that *HIP1R* mRNA expression was tightly correlated with immune-related gene sets in lung adenocarcinoma. These GSEA results suggested that *HIP1R* mRNA expression plays an important role in regulating the expression of PD-L1.

GSEA revealed that low *HIP1R* mRNA expression was closely associated with allograft rejection, inflammatory responses, IL6-JAK-STAT3, IL2-STAT5, and interferon gamma response pathways in lung adenocarcinoma. PD-L1 expression is correlated with marked expression of adaptive immune responses (CD8+ T-cells) [21]. In our study, CD8 was also included in the core enrichment gene list of HALLMARK_ALLOGRAFT_REJECTION. Previous studies have also reported that the IL6-JAK-STAT3 pathway induces PD-L1 upregulation. IL-6 is positively correlated with PD-L1 expression in human hepatocellular carcinoma (HCC) cells, and IL-6 induces PD-L1 stability through glycosylation in a HCC cell line [22]. Glioblastoma-derived IL6 is required for up-regulation of myeloid PD-L1 in glioblastoma through a STAT3-dependent mechanism [23]. Combined blockade of IL6 and PD-L1 signaling achieves synergistic antitumor immune responses in colon carcinoma and murine melanoma models [24]. PD-L1 expression is also regulated by interferon gamma signaling in a melanoma cell line [25]. GSEA suggested that HIP1R expression plays an important role in adaptive immune responses associated with PD-L1.

In the present study, no correlation was identified between HIP1R and PD-L1 protein expression, However, HIP1R mRNA expression was negatively correlated with the mRNA expression level of PD-L1 in adenocarcinoma and squamous cell carcinoma. There are several possible explanations for this discrepancy. Post-transcriptional and post-translational modifications affect the level of protein expression [26]. Proteins can have significantly different half-lives in vivo [27]. There were no cases of surgery in our study, therefore the only sample we received was a biopsy. We cannot conduct additional experiments for mRNA testing of HIP1R and PD-L1, because very little tumor tissue remains in paraffin tissue.

Patients with high HIP1R mRNA expression exhibited poor clinical outcomes in web-based mRNA data of adenocarcinoma cases; however, HIP1R mRNA expression was not correlated with OS in squamous cell carcinoma. From our IHC data, HIP1R expression was correlated with poor clinical outcomes. However, we did perform subgroup analysis according to histologic type because of our small sample size. HIP1R levels also correlate with a set of genes that reflect PD-L1-related immune pathways in GSEA analysis of adenocarcinoma cases; however, there were no statistically significant immune-related gene sets associated with HIP1R mRNA expression in squamous cell carcinoma. Currently, lung adenocarcinoma and squamous cell carcinoma are known to involve different biologic mechanisms and prognoses. Therefore, the role of HIP1R in adaptive immune responses and its effect on clinical outcomes may vary depending on the histological type.

Despite some surprising discoveries, our study has certain limitations. First, our cohort was small. We performed multivariate logistic regression and prognostic analyses on only 52 samples. However, the web-based mRNA dataset also revealed results similar to ours. These results encourage further investigations involving larger populations. Second, we used an IHC method to detect HIP1R protein expression. There is no information regarding standardization, reliability, and reproducibility of IHC staining. We used the same antibody that Wang et al. used [12]. However, Wang et al. used HIP1R antibody (16814-1-AP) in Western Blott (WB) and immunofluorescence alone. Only recently has HIP1R attracted attention in cancer research, so few studies have been done on HIP1R. Therefore, there are no antibodies that are commonly used in immunohistochemistry. In the catalog of HIP1R antibody (16814-1-AP), it can be used in IHC, immunoprecipitation (IP), WB, and ELISA. According to the manufacturer’s guidelines, this antibody was validated by western blot in HeLa cells and human liver tissue. We used a human placenta tissue as positive control as recommended. An automatic IHC staining device (Benchmark XT) may improve the reproducibility of IHC staining. The H-scoring method is widely used for immunochemical staining, and is known to have relatively high reproducibility among pathologists [28,29]. Third, we examined the protein expression of HIP1R and PD-L1 in refractory advanced NSCLC; however, the mRNA profiles of HIP1R and PD-L1 were not evaluated. Because protein expression of HIP1R did not correlate with PD-L1 expression, the relationship between HIP1R and PD-L1 mRNA expression profile is very important. To verify the results of GSEA, we should evaluate the mRNA expression profiles of HIP1R. However, the sample we have is a small biopsy, and we have already performed several immunohistochemical stainings for diagnosis and ALK and EGFR mutation tests. Therefore, currently, very little tumor tissue remains in paraffin tissue and we cannot conduct additional experiments for mRNA testing. To confirm our experiments, future research should measure the mRNA expression level of HIP1R on many samples and investigate the relationship with the PD-1 blocker and PD-L1 expression.

In conclusion, we examined the expression of HIP1R in 52 refractory NSCLC samples from patients treated with PD-1 inhibitors. HIP1R expression was an independent biomarker predicting patient response to PD-1 inhibitors. High HIP1R expression was an independent predictor of poor PFS. In addition, HIP1R mRNA expression was significantly correlated with immune-related gene sets in lung adenocarcinoma. These immune-related gene sets are known to play important roles in PD-L1 regulation. Based on our findings, HIP1R expression may be a promising predictor for the therapeutic determination of responses to anti-PD-1 treatment.

## Figures and Tables

**Figure 1 jcm-09-01425-f001:**
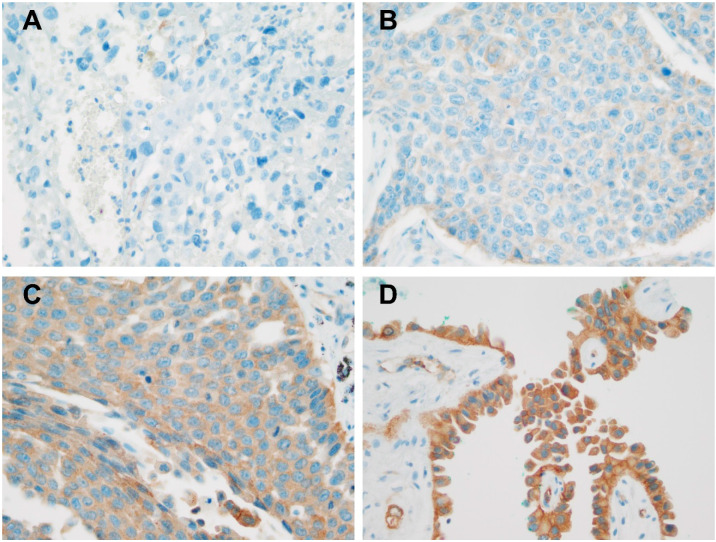
Huntingtin-interacting protein 1-related protein (HIP1R) expression in nonsmall cell carcinoma. (**A**) No staining of HIP1R, x400. (**B**) Faint HIP1R staining, X400. (**C**) Moderate HIP1R staining, X400. (**D**) Strong HIP1R staining, X400.

**Figure 2 jcm-09-01425-f002:**
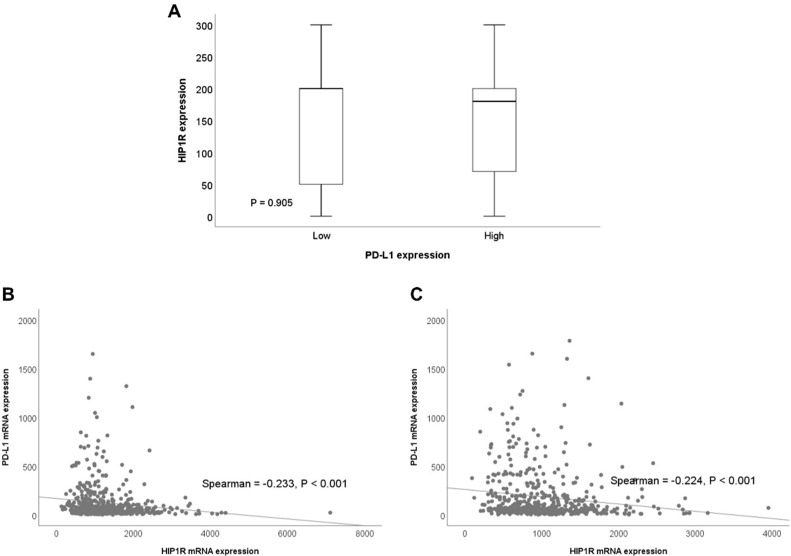
Correlation analyses involving Huntingtin-interacting protein 1-related protein (HIP1R) and programmed death-ligand 1 (PD-L1) expression. (**A**) Correlation between HIP1R and PD-L1 detected immunohistochemically. (**B**) Correlation between HIP1R and PD-L1 mRNA expression in lung adenocarcinoma. (**C**) Correlation between HIP1R and PD-L1 mRNA expression in lung squamous cell carcinoma.

**Figure 3 jcm-09-01425-f003:**
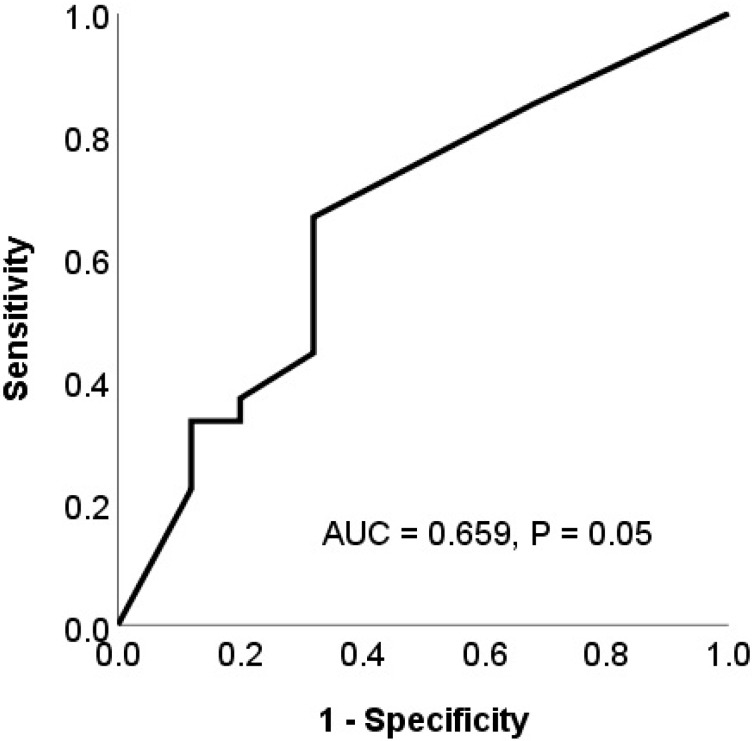
Receiver operating characteristic (ROC) and area under the curve (AUC) analysis of HIP1R expression.

**Figure 4 jcm-09-01425-f004:**
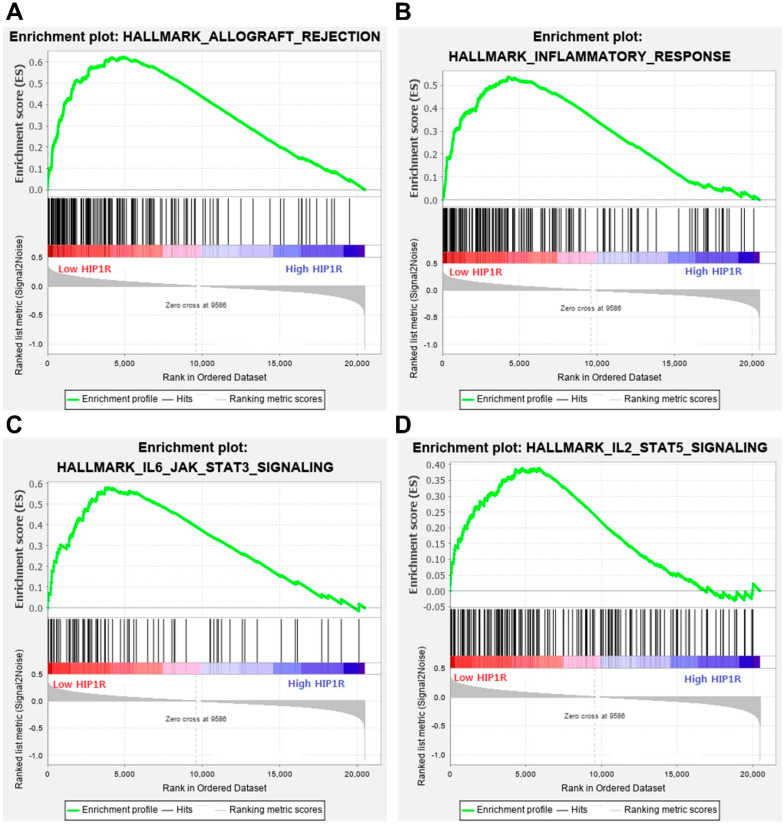
Gene set enrichment analysis (GSEA) according to HIP1R mRNA expression. (**A**) HALLMARK_ALLOGRAFT_REJECTION pathway; (**B**) HALLMARK_INFLAMMATORY_RESPONSE pathway; (**C**) HALLMARK_IL6_JAK_STAT3_SIGNALING pathway; (**D**) HALLMARK_IL2_STAT5_SIGNALING pathway.

**Figure 5 jcm-09-01425-f005:**
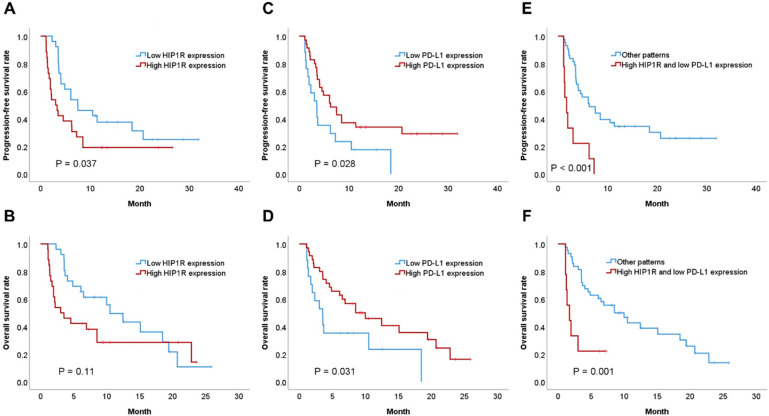
Comparison of survival rates according to HIP1R and PD-L1 expression. (**A**) Progression-free survival (PFS) and expression of HIP1R. (**B**) Overall survival (OS) and expression of HIP1R (**C**) PFS and PD-L1. (**D**) OS and PD-L1. (**E**) PFS, HIP1R, and PD-L1. (**F**) OS, HIP1R, and PD-L1.

**Table 1 jcm-09-01425-t001:** Demographic and clinical characteristics of patients.

Variable	Number (%)
Age, Median (Range) (Years)	64 (38–85)
Male Sex	43 (82.7%)
Smoking Sistory	31 (73.8%)
Histologic Subtype	
Adenocarcinoma	22 (42.3%)
Squamous Cell Carcinoma	19 (36.5%)
Pleomorphic Carcinoma	4 (7.7%)
NSCLC, NOS	7 (13.5%)
Clinical Stage at Diagnosis	
III	13 (25%)
IV	39 (75%)
Genetic Alteration Status	
EGFR-Mutated	4 (9.1%)
ALK-Rearranged	0 (0%)
Wild Type	44 (92.3%)
Type of PD-1 Blockade	
Nivolumab	27 (51.9%)
Pembrolizumab	25 (48.1%)
PD-L1 Expression	
Low (<50%)	17 (32.7%)
High (≥50%)	35 (67.3%)
Response to PD-1 Blockade	
Responder	27 (51.9%)
Nonresponder	25 (48.1%)

Smoking history was collected for 42 patients. EGFR test was performed in 44 patients. ALK test was performed in 47 patients. Abbreviations: epidermal growth factor receptor; EGFR, nonsmall-cell lung cancer—not otherwise specified; NSCLC, NOS, programmed cell death protein 1, PD-1; programmed death-ligand 1, PD-L1.

**Table 2 jcm-09-01425-t002:** Univariate and multivariate logistic regression analysis for predicting clinical response to PD-1 blockade.

	Univariate Analysis			Multivariate Analysis		
Covariate	OR	95% CI	*p*-Value ^†^	OR	95% CI	*p*-Value ^†^
Age (≥65 Years vs. <65 Years)	1.591	0.532–4.757	0.406			
Sex (Male vs. Female)	2.526	0.558–11.44	0.229			
Smoking History (+ vs. −)	3.238	0.720–14.56	0.126			
Presence of EGFR Mutation (+ vs. −)	0.222	0.021–2.330	0.210			
Type of PD-1 Blockade (Nivolumab vs. Pembrolizumab)	1.875	0.622–5.649	0.264			
PD-L1 (>50% vs. ≤50%)	4.062	1.166–14.15	0.028	4.664	1.198–18.15	0.026
HIP1R (>180 vs. ≤180)	0.235	0.074–0.751	0.015	0.209	0.060–0.731	0.014

**^†^** logistic regression analysis. Abbreviations: confidence interval; CI, epidermal growth factor receptor; EGFR, Huntingtin Interacting Protein 1 Related; HIP1R, odd ratio; OR, programmed cell death protein 1, PD-1; programmed death-ligand 1, PD-L1.

**Table 3 jcm-09-01425-t003:** Univariate and multivariate analyses of progression-free survival.

	Univariate Analysis			Multivariate Analysis		
Covariate	HR	95% CI	*p*-Value^ †^	HR	95% CI	*p*-Value ^†^
Age (≥65 Years vs. <65 Years)	1.120	0.597–2.101	0.724			
Sex (Male vs. Female)	0.553	0.251–1.218	0.141			
Smoking History (+ vs. −)	1.121	0.517–2.429	0.773			
Presence of EGFR Mutation (+ vs. −)	1.603	0.482–5.329	0.441			
Type of PD-1 Blockade (Nivolumab vs. Pembrolizumab)	1.482	0.785–2.800	0.225			
PD-L1 (>50% vs. ≤50%)	0.489	0.254–0.942	0.032	0.432	0.222–0.844	0.014
HIP1R (>180 vs. ≤180)	1.935	1.027–3.648	0.041	2.098	1.136–4.133	0.019

**^†^** Cox proportional hazards regression model. Abbreviations: CI, confidence interval; EGFR, epidermal growth factor receptor; HIP1R, Huntingtin Interacting Protein 1 Related; HR, hazard ratio; PD-1, programmed cell death protein 1; PD-L1, programmed death-ligand 1.

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
