# Peer review of "HIP1R Expression and Its Association with PD-1 Pathway Blockade Response in Refractory Advanced NonSmall Cell Lung Cancer: A Gene Set Enrichment Analysis"

_jcm, 2020, doi:10.3390/jcm9051425_

Round 1
Reviewer 1 Report
The relevance of immune-checkpoints and related cancer therapies is rapidly emerging and, as the authors say in their introduction PD-L1 is currently the most used biomarker for possible immune-checkpoint therapies.
For this reason, novel biomarkers are urgently needed as well as a better understanding of the PD-1 pathway.
In this interesting article Young at al investigate the expression levels of the protein HIP1R in a small panel of NSCLC tumours, to confirm its predictive or prognostic role in anti PD-1 therapy. The correlation analyses of H1P1R and PD-L1 expression levels have been performed on the basis of IHC staining in the available samples and using mRNA from a TCGA dataset of lung adenocarcinoma and squamous cell carcinoma. The authors didn’t found correlation from the IHC analysis but they found a negative correlation in the mRNA expression levels of HIP1R and PD-L1. Another interesting and relevant result is the role of HIP1R as an independent predictor of response to anti PD-1 therapy.
Moreover, the authors performed a Gene set enrichment analysis (GSEA) to detect genes associated with HIP1R mRNA expression, from the TCGA mRNA data set. From this analysis the authors identified 20 upregulated pathways in the low H1P1R mRNA expression group and 4 out of 20 were immune related gens with a statistical significance.
Finally, the authors precisely demonstrate that high expression of HIP1R, detected by IHC, is an independent prognostic factor of Progression Free Survival (PFS) and that high levels of HIP1R mRNA are correlated with a poor overall survival (OS) in lung adenocarcinoma.
Despite the considerable interest of the reported results, it’s my opinion that the article could be improved and require some additional information and correction that I will report in this revision.
- As the authors say in lane 253 of the discussion one of the main limits and probably the greatest one of this study is the small number of samples. If possible, it could be fine to increase the numbers of the patients in the study to have an increase in the statistical relevance.
- In lane 80 I really appreciate that the authors correctly indicate all the information regarding the anti HI1PR antibody, nevertheless in the discussion (lanes 255 – 257) they say that there are no information’s about the standardization and the reproducibility of the IHC staining. The authors indicate that the antibody used is the same of the Wang et al article, but in that article the antibody is used only for Western Blott or for Immunofluorescence.
The standardization of a new IHC protocol is extremely important but, in my opinion, it needs to be improved confirming it using an already validated IHC antibody.
- Lane 90 (Paragraph 2.3). Please indicate which is the antibody used for PD-L1 IHC staining and the dilution used. Moreover, it will be nice to show some images of the immunoistochemical staining.
- Lane 144 (Paragraph 3.2). The obtained results are clear and well presented, nevertheless in my opinion could be improved evaluating if the negative correlation found in the mRNA TCGA data set, is also present in the 48 tissue samples available to authors. The RNA extracted from Formalin Fixed Paraffine Embedded (FFPE) tissue samples could be analysed quantitatively by Real Time PCR.
- Lane 176 (Paragraph 3.4). The article could considerably improve its interest with an analysis of the expression levels of the four (or a couple of it) immune related genes revealed by GSEA also in the 48 samples studied by the authors. The levels of expression could be analysed by IHC with specific antibodies and/or by Real Time PCR on the mRNA extracted by FFPE.
- In lane 200, please replace inferior with lower.
- Lane 205 (Paragraph 3.5). Please clarify to this reviewer if it’s possible to have information about the OS of the 48 patients analysed in the study and if there is a correlation with HIP1R expression levels (by IHC or by mRNA levels from FFPE samples).
- Lane 241. Please remove the sentence “It is difficult to avoid errors and noise in either protein or mRNA experiments”. Usually the laboratory where these experiments are performed frequently work hard and precisely to standardize protocols to have clear and reproducible (three times or more) results without or with a very low possibility of error.
- Lane 268. Please replace pretreatment with refractory.
Author Response
# reviewer 1
The relevance of immune-checkpoints and related cancer therapies is rapidly emerging and, as the authors say in their introduction PD-L1 is currently the most used biomarker for possible immune-checkpoint therapies.
For this reason, novel biomarkers are urgently needed as well as a better understanding of the PD-1 pathway.
In this interesting article Young at al investigate the expression levels of the protein HIP1R in a small panel of NSCLC tumours, to confirm its predictive or prognostic role in anti PD-1 therapy. The correlation analyses of H1P1R and PD-L1 expression levels have been performed on the basis of IHC staining in the available samples and using mRNA from a TCGA dataset of lung adenocarcinoma and squamous cell carcinoma. The authors didn’t found correlation from the IHC analysis but they found a negative correlation in the mRNA expression levels of HIP1R and PD-L1. Another interesting and relevant result is the role of HIP1R as an independent predictor of response to anti PD-1 therapy.
Moreover, the authors performed a Gene set enrichment analysis (GSEA) to detect genes associated with HIP1R mRNA expression, from the TCGA mRNA data set. From this analysis the authors identified 20 upregulated pathways in the low H1P1R mRNA expression group and 4 out of 20 were immune related gens with a statistical significance.
Finally, the authors precisely demonstrate that high expression of HIP1R, detected by IHC, is an independent prognostic factor of Progression Free Survival (PFS) and that high levels of HIP1R mRNA are correlated with a poor overall survival (OS) in lung adenocarcinoma.
Despite the considerable interest of the reported results, it’s my opinion that the article could be improved and require some additional information and correction that I will report in this revision.
- As the authors say in lane 253 of the discussion one of the main limits and probably the greatest one of this study is the small number of samples. If possible, it could be fine to increase the numbers of the patients in the study to have an increase in the statistical relevance.
Response: The patients who received the PD-1 blocker were examined again, but only 4 were added. We modified the results section, figures and tables based on a total of 52 samples. Similar results were obtained with the 48 samples.
Modified results
“3.1. Patient demographics
Detailed patient and tumor characteristics are summarized in Table 1. Forty-six tissues were collected from lung lesions, and six were obtained from metastatic sites. Twenty-seven (51.9%) patients had been treated with nivolumab, and 25 (48.1%) patients received pembrolizumab. All patients were refractory to conventional treatments such as chemotherapy, radiotherapy or target therapy. Therefore, they received PD-1 inhibitor as a second line or later setting. Twenty-seven (51.9%) patients were classified as responders and 25 (48.1%) were classified as non-responders. Four patients were treated with epidermal growth factor receptor (EGFR) inhibitors before PD-1 inhibitor administration. Patients with anaplastic lymphoma kinase (ALK) fusion was not identified in the present study.
3.2. Relationships between HIP1R and PD-L1 analyzed by IHC and mRNA expression
We performed correlation analysis of HIP1R and PD-L1 expression using IHC techniques. There was no statistically significant correlation between HIP1R and PD-L1 expression (p = 0.905, Figure 2A).
3.3. Associations involving HIP1R, PD-L1, clinicopathologic parameters, and response to PD-1 inhibitors
ROC analysis was performed to determine the cut-off value for HIP1R expression. Cut-off was determined as the value corresponding to the maximum joint sensitivity and specificity of the ROC curve. The area under the curve (AUC) was 0.659 for the expression of HIP1R, and the cut-off value was 180 (66% sensitivity and 68% specificity, Figure 3).
We explored the predictive capacity of HIP1R, PD-L1, and clinicopathologic factors in terms of responses to PD-1 inhibition. By univariate analysis, the expression of HIP1R was found to be a predictor of the response to anti-PD-1 therapy (OR) = 0.235, p = 0.015; Table 2). PD-L1 expression was also found to be a predictor of the response to anti-PD-1 therapy (OR = 4.062, p = 0.028; Table 2). By multivariate analysis, the expression of HIP1R was an independent predictor of anti-PD-1 therapy response (OR = 0.209, p = 0.014, Table 2).
3.5. Prognostic significance of HIP1R and PD-L1
Patients with high HIP1R expression had inferior PFS than patients with low HIP1R expression (p = 0.037, Figure 5A). Patients with high HIP1R expression also showed an inferior OS than patients with low HIP1R expression, however the statistical significance was not reached (p = 0.11, Figure 5B). Patients with high PD-L1 expression had superior PFS or OS than patients with low PD-L1 expression (p = 0.028, Figure 5C and p = 0.031, Figure 5D, respectively). Furthermore, patients with high HIP1R expression and low PD-L1 expression had lower PFS or OS than patients with other expression patterns (p < 0.001, Figure 5E and p = 0.001, Figure 5F, respectively). In multivariate analysis, high HIP1R expression was an independent prognostic factor for PFS (HR = 2.098, p = 0.019, Table 3).
- In lane 80 I really appreciate that the authors correctly indicate all the information regarding the anti HI1PR antibody, nevertheless in the discussion (lanes 255 – 257) they say that there are no information’s about the standardization and the reproducibility of the IHC staining. The authors indicate that the antibody used is the same of the Wang et alarticle, but in that article the antibody is used only for Western Blott or for Immunofluorescence.
The standardization of a new IHC protocol is extremely important but, in my opinion, it needs to be improved confirming it using an already validated IHC antibody.
Response: We also agree that Wang et al used HIP1R antibody (16814-1-AP) in Western Blott and Immunofluorescence. In the catalog of HIP1R antibody (16814-1-AP), it was tested by immunohistochemistry (IHC), Immunoprecipitation (IP), Western Blott (WB) and ELISA. It is said that it can be used in IHC, IP, WB, ELISA. We used a human placenta tissue as positive control as catalog recommended. Only recently has HIP1R attracted attention in cancer research, so few studies have been done on HIP1R. Therefore, there are no antibodies that are commonly used in immunohistochemistry. We added this comments in discussion section.
Lane 257-268:” Second, we used an IHC method to detect HIP1R protein expression. There is no information regarding standardization, reliability, and reproducibility of IHC staining. We used the same antibody that Wang et al. used[12]. However, Wang et al used HIP1R antibody (16814-1-AP) in Western Blott (WB) and Immunofluorescence alone. Only recently has HIP1R attracted attention in cancer research, so few studies have been done on HIP1R. Therefore, there are no antibodies that are commonly used in immunohistochemistry. In the catalog of HIP1R antibody (16814-1-AP), it can be used in IHC, immunoprecipitation (IP), WB, ELISA. According to the manufacturer's guidelines, this antibody was validated by western blot in HeLa cells and human liver tissue. We used a human placenta tissue as positive control as recommended. An automatic IHC staining device (Benchmark XT) may improve the reproducibility of IHC staining. The H-scoring method is widely used for immunochemical staining and is known to have relatively high reproducibility among pathologists”
- Lane 90 (Paragraph 2.3). Please indicate which is the antibody used for PD-L1 IHC staining and the dilution used. Moreover, it will be nice to show some images of the immunoistochemical staining.
Response: Two PD-L1 antibodies (clone name SP263 or 22C3) were used to detect PD-L1 expression. Sp263 was a companion diagnostic assay for OPDIVO® (nivolumab) and 22c3 was a companion diagnostic assay for KEYTRUDA® (pembrolizumab). Two PD-L1 test used prediluted antibody (ready to use) according to the protocol. We added this comments in materials and methods section. We added images of the PD-L1 immunohistochemical staining (supplementary figure 1 and 2).
Lane 91-98: Two PD-L1 antibodies (clone name SP263 or 22C3) were used to detect PD-L1 expression. Sp263 was a companion diagnostic assay for OPDIVO® (nivolumab) and 22c3 was a companion diagnostic assay for KEYTRUDA® (pembrolizumab). We performed SP263 and/or 22C3 assays prior to PD-1 inhibitor treatment for all NSCLC patients. Thirteen (25%) of the 52 specimens were tested for both SP263 and 22C3, 27 (51.9%) for only SP263, and 12 (23.1%) for only 22C3. Two PD-L1 test used prediluted antibody (ready to use) according to the protocol. The SP263 assay was performed using a VENTANA BenchMark ULTRA instrument and the 22C3 assay was conducted using the Dako Link-48 platform, as recommended by the manufacturers [16].
- Lane 144 (Paragraph 3.2). The obtained results are clear and well presented, nevertheless in my opinion could be improved evaluating if the negative correlation found in the mRNA TCGA data set, is also present in the 48 tissue samples available to authors. The RNA extracted from Formalin Fixed Paraffine Embedded (FFPE) tissue samples could be analysed quantitatively by Real Time PCR.
Response: We also agree that better results may be achieved by adding mRNA data from the patient group. Our patient group is advanced NSCLC patients. There were no cases of surgery, therefore the only sample we received was a bronchoscopic or CT-guided biopsy. Lung biopsy samples are difficult to collect, so they are generally smaller in size than samples from other organs.
We performed routine immunohistochemistry (TTF1, NapsinA and p40) for histologic subclassification. We also performed immunohistochemistry for PD-L1 (sp263 or 22c3) and ALK gene translocation. We conducted an EGFR mutation test in most patients. Therefore, currently, very little tumor tissue remains in paraffin tissue. We cutted paraffin blocks of several samples for mRNA collection, but in this case, there was a risk of completely losing the tumor tissue on the paraffin tissue. Due to the current hospital policy, minimal tumor paraffin blocks must be stored because patients may want to go to another hospital. Therefore, we cannot conduct additional experiments for mRNA testing. Please consider the realistic problems for a small biopsy sample. This comments was added in discussion section.
Lane 240-243: “There were no cases of surgery in our study, therefore the only sample we received was a biopsy. We cannot conduct additional experiments for mRNA testing of HIP1R and PD-L1 because very little tumor tissue remains in paraffin tissue.”
- Lane 176 (Paragraph 3.4). The article could considerably improve its interest with an analysis of the expression levels of the four (or a couple of it) immune related genes revealed by GSEA also in the 48 samples studied by the authors. The levels of expression could be analysed by IHC with specific antibodies and/or by Real Time PCR on the mRNA extracted by FFPE.
Response: We also agree that better results may be achieved by adding mRNA or IHC data of four immune-related gene from the patient group. The amount of remaining sample is too small, so additional experimentation is difficult. Please consider the realistic problems for a small biopsy sample
- In lane 200, please replace inferior with lower.
Response: In lane 200, we replace inferior with lower.
Lane 200: “Furthermore, patients with high HIP1R expression and low PD-L1 expression had lower PFS or OS than patients with other expression patterns (p < 0.001, Figure 5E and p = 0.001, Figure 5F, respectively).”
- Lane 205 (Paragraph 3.5). Please clarify to this reviewer if it’s possible to have information about the OS of the 48 patients analysed in the study and if there is a correlation with HIP1R expression levels (by IHC or by mRNA levels from FFPE samples).
Response: We added the information about the OS.
Lane 195-202: “Patients with high HIP1R expression had inferior PFS than patients with low HIP1R expression (p = 0.037, Figure 5A). Patients with high HIP1R expression also showed an inferior OS than patients with low HIP1R expression, however the statistical significance was not reached (p = 0.11, Figure 5B). Patients with high PD-L1 expression had superior PFS or OS than patients with low PD-L1 expression (p = 0.028, Figure 5C and p = 0.031, Figure 5D, respectively). Furthermore, patients with high HIP1R expression and low PD-L1 expression had lower PFS or OS than patients with other expression patterns (p < 0.001, Figure 5E and p = 0.001, Figure 5F, respectively). In multivariate analysis, high HIP1R expression was an independent prognostic factor for PFS (HR = 2.098, p = 0.019, Table 3).”
- Lane 241. Please remove the sentence “It is difficult to avoid errors and noise in either protein or mRNA experiments”. Usually the laboratory where these experiments are performed frequently work hard and precisely to standardize protocols to have clear and reproducible (three times or more) results without or with a very low possibility of error.
Response: We remove the sentence “It is difficult to avoid errors and noise in either protein or mRNA experiments”
- Lane 268. Please replace pretreatment with refractory.
Response: In lane 264, we replace pretreatment with refractory.
Lane 264: “In conclusion, we examined the expression of HIP1R in 52 refractory NSCLC samples from patients treated with PD-1 inhibitors.”
Reviewer 2 Report
Koh YW and collaborator try to show the role of HIP1R as possible biomarker in anti-PD-1 treatment in lung cancer. Indeed, HIP1R plays an important role in the regulation PD-L1 degradation nevertheless, relationships involving between HIP1R and PDL-1 in lung cancer is not yet clear.
The idea is interesting but the data presented are not sufficient to support the hypothesis, they need to take another experiments to increase paper significance and quality.
- The authors did not show statistically significant correlation between HIP1R and PD-L1 expression through an IHC analysis using samples derived from retrospectively selected 48 advanced NSCLC patients. Moreover, they observed from the TCGA dataset that HIP1R mRNA expression levels were negatively correlated with PD-L1 mRNA levels in adenocarcinoma and squamous cell carcinoma. The authors should confirm the TCGA data, analyzing le mRNA level of HIP1R and PD-L1 in the 48 patient samples.
- The authors performed GSEA to identify gene sets associated with HIP1R mRNA expression in the TCGA mRNA data of lung adenocarcinoma and lung squamous cell carcinoma cases. To support this data, the authors should develop RNA-seq experiment using RNA extracted from 48 patient samples.
- GSEA showed that 4 immune-related pathways were upregulated in the low HIP1R mRNA expression group. The authors, should confirm these results through RNA-seq analysis. Subsequently, they should investigate if this pathway are differently modulate in patients classified as responders or non-responders after PD-1 treatment.
Author Response
# reviewer 2
Koh YW and collaborator try to show the role of HIP1R as possible biomarker in anti-PD-1 treatment in lung cancer. Indeed, HIP1R plays an important role in the regulation PD-L1 degradation nevertheless, relationships involving between HIP1R and PDL-1 in lung cancer is not yet clear.
The idea is interesting but the data presented are not sufficient to support the hypothesis, they need to take another experiments to increase paper significance and quality.
- The authors did not show statistically significant correlation between HIP1R and PD-L1 expression through an IHC analysis using samples derived from retrospectively selected 48 advanced NSCLC patients. Moreover, they observed from the TCGA dataset that HIP1R mRNA expression levels were negatively correlated with PD-L1 mRNA levels in adenocarcinoma and squamous cell carcinoma. The authors should confirm the TCGA data, analyzing le mRNA level of HIP1R and PD-L1 in the 48 patient samples.
Response: We also agree that better results may be achieved by adding mRNA data from the patient group. Our patient group is advanced NSCLC patients. There were no cases of surgery, therefore the only sample we received was a bronchoscopic or CT-guided biopsy. Lung biopsy samples are difficult to collect, so they are generally smaller in size than samples from other organs. We performed routine immunohistochemistry (TTF1, NapsinA and p40) for histologic subclassification. We also performed immunohistochemistry for PD-L1 (sp263 or 22c3) and ALK gene translocation. We conducted an EGFR mutation test in most patients. Therefore, currently, very little tumor tissue remains in paraffin tissue. We cutted paraffin blocks of several samples for mRNA collection, but in this case, there was a risk of completely losing the tumor tissue on the paraffin tissue. Due to the current hospital policy, minimal tumor paraffin blocks must be stored because patients may want to go to another hospital. Therefore, we cannot conduct additional experiments for mRNA testing. Please consider the realistic problems for a small biopsy sample. This comments was added in discussion section.
Lane 240-243: “There were no cases of surgery in our study, therefore the only sample we received was a biopsy. We cannot conduct additional experiments for mRNA testing of HIP1R and PD-L1 because very little tumor tissue remains in paraffin tissue.”
- The authors performed GSEA to identify gene sets associated with HIP1R mRNA expression in the TCGA mRNA data of lung adenocarcinoma and lung squamous cell carcinoma cases. To support this data, the authors should develop RNA-seq experiment using RNA extracted from 48 patient samples.
Response: We also agree that better results may be achieved by adding RNA-seq data from the patient group. The amount of remaining sample is too small, so additional experimentation is difficult. Please consider the realistic problems for a small biopsy sample
- GSEA showed that 4 immune-related pathways were upregulated in the low HIP1R mRNA expression group. The authors, should confirm these results through RNA-seq analysis. Subsequently, they should investigate if this pathway are differently modulate in patients classified as responders or non-responders after PD-1 treatment.
Response: If there is RNA-seq data according to the response group in our sample, it must be an important result. However, the amount of remaining sample is too small, so additional experimentation is difficult. Please consider the realistic problems for a small biopsy sample.
Round 2
Reviewer 1 Report
Dear Authors
This Reviewer appreciated your corrections and the improvement of your article. However, I would have few other suggestions and notations to make:
- Referring to figure 1 and to supplementary figures is it possible to insert the magnification level in all the IHC images?
- Lane 257-268, referring to the human placenta tissue used as positive control, it will be nice to show an exemplificative image in the supplementary section. In this way also, the methods section about the IHC will be strongly increased.
Finally, I understand the problems of the small biopsies samples, but in my opinion the suggestion about the analyses of the mRNA levels in your sample and about the analyses (mRNA or Protein) on some of the genes revealed by GSEA could really confirm your data. For this reason, I suggest you to plan, in future studies, this kind of analyses and expose it in the discussion section as future perspectives.
Moreover, in the recommendations for authors section, I did not change (and I will not change) the point: “are the conclusions supported by the results?”, because I still think that the analyses suggested could contribute to the validation of all the results presented by the authors and increase the interest of the article.
Author Response
Dear Authors
This Reviewer appreciated your corrections and the improvement of your article. However, I would have few other suggestions and notations to make:
- Referring to figure 1 and to supplementary figures is it possible to insert the magnification level in all the IHC images?
Response: We added the magnification level in all the IHC figure legends.
Lane 88-89 “Figure 1. HIP1R expression in non-small cell carcinoma. (A) No staining of HIP1R, x400. (B) Faint HIP1R staining, X400. (C) Moderate HIP1R staining, X400. (D) Strong HIP1R staining, X400.”
Supplementary Figure Legends
Figure S1. HIP1R expression in positive control. Positive HIP1R expression in placental tissue, X400
Figure S2. PD-L1 sp263 expression in non-small cell carcinoma. (A) No staining of PD-L1 sp263, x400. (B) Faint PD-L1 sp263 staining, x400. (C) Moderate PD-L1 sp263 staining, x400. (D) Strong PD-L1 sp263 staining, x400.
Figure S3. PD-L1 22c3 expression in non-small cell carcinoma. (A) No staining of PD-L1 22c3, x400. (B) Faint PD-L1 22c3 staining, x400. (C) Moderate PD-L1 22c3 staining, x400. (D) Strong PD-L1 22c3 staining, x400.
- Lane 257-268, referring to the human placenta tissue used as positive control, it will be nice to show an exemplificative image in the supplementary section. In this way also, the methods section about the IHC will be strongly increased.
Response: We added the exemplificative image of HIP1R expression of human placental tissue in the supplementary section.
Lane 81-82 “We used a human placenta tissue as positive control according to the manufacturer's recommendations (Figure S1).”
- Finally, I understand the problems of the small biopsies samples, but in my opinion the suggestion about the analyses of the mRNA levels in your sample and about the analyses (mRNA or Protein) on some of the genes revealed by GSEA could really confirm your data. For this reason, I suggest you to plan, in future studies, this kind of analyses and expose it in the discussion section as future perspectives.
Response: To verify our experiment, we described that HIP1R mRNA test should be performed in future research in the discussion section. If possible, we would like to verify our results by conducting HIP1R mRNA test in the future.
Lane 272-281 “Third, we examined the protein expression of HIP1R and PD-L1 in refractory advanced NSCLC, however the mRNA profiles of HIP1R and PD-L1 were not evaluated. Because protein expression of HIP1R did not correlate with PD-L1 expression, the relationship between HIP1R and PD-L1 mRNA expression profile is very important. To verify the results of GSEA, we should evaluate the mRNA expression profiles of HIP1R. However, the sample we have is a small biopsy, and we have already performed several immunohistochemical stainings for diagnosis and ALK and EGFR mutation tests. Therefore, currently, very little tumor tissue remains in paraffin tissue and we cannot conduct additional experiments for mRNA testing. To confirm our experiments, future research should measure the mRNA expression level of HIP1R on many samples and investigate the relationship with the PD-1 blocker and PD-L1 expression.”
- Moreover, in the recommendations for authors section, I did not change (and I will not change) the point: “are the conclusions supported by the results?”, because I still think that the analyses suggested could contribute to the validation of all the results presented by the authors and increase the interest of the article.
Response: We also agree that the analysis proposed by the reviewer (mRNA test) is essential to verify our results and increase the interest of the article. Therefore, future studies should measure mRNA expression levels of HIP1R in large-scale cohort and investigate the relationship between PD-1 blockers and PD-L1 expression. We described that HIP1R mRNA test should be performed in future research in the discussion section.
Lane 272-281 “Third, we examined the protein expression of HIP1R and PD-L1 in refractory advanced NSCLC, however the mRNA profiles of HIP1R and PD-L1 were not evaluated. Because protein expression of HIP1R did not correlate with PD-L1 expression, the relationship between HIP1R and PD-L1 mRNA expression profile is very important. To verify the results of GSEA, we should evaluate the mRNA expression profiles of HIP1R. However, the sample we have is a small biopsy, and we have already performed several immunohistochemical stainings for diagnosis and ALK and EGFR mutation tests. Therefore, currently, very little tumor tissue remains in paraffin tissue and we cannot conduct additional experiments for mRNA testing. To confirm our experiments, future research should measure the mRNA expression level of HIP1R on many samples and investigate the relationship with the PD-1 blocker and PD-L1 expression.”